# *Drosophila melanogaster*: A Model Organism in Muscular Dystrophy Studies

**DOI:** 10.3390/ijms26041459

**Published:** 2025-02-10

**Authors:** Yu Zhao, Yujie Wang, Ayibota Tulehalede, Zhu Meng, Lizhong Xu, Huashuai Bai, Junhui Sha, Wei Xie, Junhua Geng

**Affiliations:** Key Laboratory of Developmental Genes and Human Disease, School of Life Science and Technology, Nanjing 210096, China220223584@seu.edu.cn (A.T.);

**Keywords:** muscular dystrophy, muscle degeneration, *Drosophila melanogaster*

## Abstract

Muscular dystrophy is a group of complicated, genetically heterogeneous disorders characterized by progressive muscle weakness and degeneration. Due to the intricate nature, understanding the molecular mechanisms underlying muscular dystrophy presents significant challenges. *Drosophila*, as a versatile and genetically tractable model organism, offers substantial advantages in muscular dystrophy research. In the present review, we summarize the application of *Drosophila* in studying various types of muscular dystrophy, highlighting the insights gained through genetic manipulations, disease modeling, and the exploration of molecular pathways. *Drosophila* serves as a powerful system for understanding disease progression, exploring the roles of key genes in muscle function and pathology, and identifying novel therapeutic targets. The review highlights the significant role of *Drosophila* in advancing our understanding of muscular dystrophy.

## 1. Introduction

Muscular dystrophy (MD) refers to a group of inherited disorders characterized by progressive muscle weakness and atrophy. These conditions arise from structural or functional defects in muscle fibers, leading to gradual muscle degeneration and then a loss of function [1,2]. MD is often accompanied by fibrosis and fat infiltration in muscle tissue, resulting in severe motor impairments and systemic weakness in advanced stages [3,4]. Based on genetic patterns and clinical manifestations, MD can be categorized into various subtypes, including Duchenne muscular dystrophy (DMD), Becker muscular dystrophy (BMD), Laminin-related muscular dystrophy (LRMD), limb-girdle muscular dystrophy (LGMD), Fukuyama congenital muscular dystrophy (FCMD), Ullrich muscular dystrophy, and dystroglycanopathies [4,5,6,7,8,9].

Recent advances in genomics and molecular biology have provided significant insights into the pathogenesis of MD [10,11]. The primary causes of MD involve mutations in genes encoding key proteins, such as dystrophin and dystroglycan, which are essential for stabilizing the muscle membrane or glycosyltransferase-like protein O-mannosyltransferases (POMTs) and fukutin, which are crucial for dystroglycan glycosylation [12,13,14,15,16]. The lack of these molecules disrupts the interactions between muscle cells and the extracellular matrix, resulting in defects in muscle development, pathological changes, and instability in muscle structure. Understanding the genetic and biochemical basis of MD not only provides insights into muscle biology but also opens avenues to develop targeted therapies for MD.

In the present review, we discuss the unique potential and current applications of *Drosophila* in MD research. We concentrate on utilizing the *Drosophila* model for obtaining a more profound comprehension of the pathogenesis underlying MD, validating the key genes and molecular pathways, and investigating the functions and pathological implications of these molecules in MD.

## 2. The Advantages of *Drosophila* in MD Research

*Drosophila* has emerged as a powerful model for studying MD due to its high genetic and functional conservation with humans. Approximately 60% of the *Drosophila* genome is shared with humans, and about 75% of human-disease-related genes have orthologs in *Drosophila* [17,18]. For example, Dystroglycan, which links the intracellular cytoskeleton to the extracellular matrix, exhibits remarkable structural and functional conservation between *Drosophila* and mammals, and its glycosylation is critical for maintaining muscle integrity [4,13,19,20,21,22,23,24,25]. Moreover, *rotated abdomen* and *twisted*, *Drosophila* homologs of glycosyltransferase genes, share functional similarities with their mammalian counterparts [23]. Mutations in the laminin gene, a crucial component of the extracellular matrix that interacts with dystroglycan, have been observed to induce muscle degeneration in *Drosophila* [26,27]. The degenerative process closely mirrors the pathological traits of muscular dystrophy (MD), highlighting the laminin structural and functional parallels between *Drosophila* and vertebrates.

The simplicity of the *Drosophila* muscle system, along with its genetic manipulability, provides a powerful platform for dissecting the molecular and cellular mechanisms underlying MD. Each mature muscle in *Drosophila* comprises a single muscle fiber, and the muscles within each half-segment exhibit similar physiological and contractile properties, making it an ideal system for studying muscle function and dysfunction associated with MD [28]. Mutations in genes associated with MD exhibit detectable defects in the muscle structure of *Drosophila* larvae (Figure 1), which provides a powerful platform for investigating the pathogenesis of the disease and identifying potential therapeutic targets.

*Drosophila* serves as an excellent genetically tractable model for studying human diseases, thereby aiding in the understanding of the pathomechanisms of DM. Traditional methods, such as P-element insertion and excision [29,30], are used to induce mutants that phenocopy the phenotypes observed in DM patients. Additionally, ethyl-methane-sulfonate (EMS) mutagenesis introduces missense mutations within the coding region of genes, such as col4a1 [31], resulting in severe myopathic phenotypes. Moreover, the CRISPR/Cas9 system, when adapted for use in *Drosophila*, enables precise genomic modifications, which facilitates the creation of disease models and the study of human genes [32]. Additionally, *Drosophila* offers a powerful set of genetic tools, including the Gal4/UAS system, LexA/LexAop, and QF/QUAS, enabling the precise control of gene expression in specific tissues or developmental stages [33,34]. The expression of disease-associated repeat tracts, such as (CCUG)_106_ and (CTG)_480_, in muscle and retinal cells leads to the formation of the ribonuclear foci and mis-splicing of genes implicated in DM pathology [35,36,37]. These binary expression systems, along with RNAi and FLP/FRT recombination systems [17,38], facilitate targeted gene manipulation. Through transgenic RNA interference, van der Plas et al. examined the roles of dystrophin isoforms in *Drosophila* muscle [39], while Shcherbata et al. dissected muscle and neuronal disorders in a *Drosophila* model of MD [40]. Furthermore, utilizing these model flies in genetic screens and functional assays, Garcia-Lopez et al. identified new components of the pathogenesis pathway and chemical suppressors of DM-like phenotypes [37].

Together, these tools make *Drosophila* an invaluable model for genetic research and disease modeling. The use of model organisms, such as *Drosophila*, has played a crucial role in elucidating disease mechanisms, exploring potential therapeutic strategies, and offering hope to individuals living with these conditions. Building on the unique advantages of *Drosophila*, this review delves into its use for modeling specific forms of MD.

## 3. Modeling Muscular Dystrophy in *Drosophila*

### 3.1. Duchenne Muscular Dystrophy

Duchenne muscular dystrophy (DMD) is a severe, X-linked pediatric disorder. Mutations in the *dystrophin* gene that disrupt its reading frame or introduce a premature stop codon, causing rapid mobility loss and premature death [41,42]. There is a single *dystrophin* gene located on the third chromosome of *Drosophila*. This gene produces several dystrophin isoforms that are homologous to human ones. Specifically, DLP1, DLP2, DLP3, Dp205, and Dp186 in *Drosophila* are homologous to Dp427, Dp260, Dp140, Dp116, and Dp71 in human, respectively [43]. Reducing the expression of pan-dystrophin in *Drosophila* led to muscle rupture, absence, or detachment from tendon cell attachment sites, mimicking the muscle degeneration shown in DMD. Despite this, embryonic muscle development showed no defects, resembling the normal muscle condition observed at birth in patients [39]. Although *dystrophin* knockdown in tendon cells did not impact muscle integrity, it caused progressive climbing defects and severe muscle degeneration in adult muscle fibers [40]. The reduction in Dp117, a truncated form of dystrophin in *Drosophila*, caused disorganized myofilaments and necrosis manifested by swollen sarcoplasmic reticulum in late pupae [39] and age-dependent cardiac defects [29], resembling dilated cardiomyopathy in DMD.

Becker muscular dystrophy (BMD) is another X-linked recessive disorder caused by mutations in the *dystrophin* gene, characterized by the progressive weakness of the muscles in the legs and pelvis. Unlike DMD, which is caused by a complete loss of dystrophin, BMD results from a partial reduction in the dystrophin protein and is, therefore, considered a milder form of DMD. The symptoms of BMD are similar to those of DMD but less severe, typically manifesting in late childhood or early adulthood. Early symptoms manifest as muscle cramps following physical activity, along with challenges in walking and running [43]. *Drosophila* models for BMD are often generated by knocking down the *dystrophin* to reduce the expression of dystrophin, allowing for the investigation of various biological processes related to BMD.

### 3.2. Myotonic Dystrophy

Myotonic dystrophy (DM) is a genetic disorder characterized by muscle weakness and myotonia. Myotonic dystrophy type 1 (DM1) and type 2 (DM2) are inherited disorders that impact various systems, including skeletal and smooth muscles, the heart, brain, eyes, and other organs [44]. Noncoding microsatellite repeat expansions, such as (CTG) expansions in DMPK and (CCTG) expansions in CNBP, are examples of the toxic RNA gain-of-function mechanism in MD, underscoring that splicing defects are directly linked to the pathology of MD [17,35,36,37,45]. For example, expressing pristine CTG repeats in the 3′ UTR region of the DMPK gene did not lead to significant locomotor defects or a shortened lifespan. In contrast, expressing 480 interrupted CTG repeats [(iCUG)480] in muscle and eye tissues resulted in age-dependent degeneration and the accumulation of RNA foci [36,37]. These foci co-localized with muscleblind (Mbl) and were accompanied by the stable expression of CELF1 in *Drosophila*, mimicking the characteristics observed in DM1 patients [46,47]. The stabilization of Bru3 also resulted in the downregulation of sarcomeric proteins [48], while the loss of MBNL1/Mbl led to the upregulation of its target genes, such as *GJA1* and *CACNA1C*, which are associated with conduction defects and arrhythmias/sudden death, respectively [49,50].

DM2, similar to DM1, results from unstable, noncoding (CCTG)_n_ repeat expansions in the *CNBP* gene, which encodes a CCHC-type zinc finger protein. Transgenic flies expressing (CCUG) repeats display RNA foci formation and the mis-splicing of MBNL1-dependent transcripts, making them ideal models for investigating DM2 [50]. Overall, *Drosophila* has demonstrated its value as a model system for studying both DM1 and DM2, providing insights into the molecular pathways, splicing defects, and tissue-specific toxicities associated with these diseases.

### 3.3. Facioscapulohumeral Muscular Dystrophy

Facioscapulohumeral muscular dystrophy (FSHD) ranks as the third most prevalent muscular dystrophy, following DMD and myotonic dystrophy. The disease is typically characterized by weakness in the facial muscles and the proximal parts of the arms, along with a condition known as winged scapula. This is often followed by weakness in the muscles responsible for foot dorsiflexion and the hip girdle. Truncal muscles, including the paraspinal and abdominal muscles, are also impaired to varying degrees [51,52]. Notably, asymmetric involvement is a common characteristic and is often markedly pronounced

FSHD is associated with epigenetic changes in the 4q35 region of the chromosome 4, leading to the overexpression of the DUX4 gene [53]. Limited by the low endogenous expression of DUX4 and its cytotoxic effects upon overexpression in somatic cells, in vivo studies of DUX4 have been challenging. The UAS/GAL4 binary system has been used to establish a *Drosophila* model with the overexpression of DUX4-fl for investigating the pathogenesis of FSHD [54]. Additionally, *FRG1* (FSHD region gene 1) is also linked to FSHD, and its overexpression causes muscle-dystrophy-like defects. FRG1 interacts with and inhibits the activity of Suv4-20h1, a histone methyltransferase, thereby suppressing muscle development [55].

A typical feature of FSHD is the shortening of the D4Z4 repeat sequence in the 4q35 region of chromosome 4, which is believed to lead to the loss of heterochromatinization and promote the overexpression of FSHD-associated genes. The prevailing model posits that the shortening of the D4Z4 repeat sequence causes a loss of heterochromatinization, similar to the position-effect variegation (PEV) observed in *Drosophila*. Moreover, the acetylation levels of histone H4 in the non-repetitive regions adjacent to the D4Z4 region resembled those typically observed in unexpressed euchromatin, rather than in constitutive heterochromatin [56]. It is quite possible that there are other changes in epigenetic modification to upregulate the transcription of the gene involved in muscular structure and function. Equally, polymorphic D4Z4 repeat is located on chromosome 10q26, but FSHD-sized D4Z4 repeats on 10q have never been associated with disease [57]. The findings indicate that the molecular underpinnings of FSHD could be more intricate than previously conceived, emphasizing the importance of employing a model organism, such as *Drosophila*, for the exploration of novel hypotheses.

### 3.4. Limb-Girdle Muscular Dystrophy

Limb-girdle muscular dystrophies (LGMDs) represent a group of rare and genetically diverse disorders, characterized by progressive muscle weakness and atrophy, primarily affecting the pelvic and shoulder girdle muscles. Traditionally, LGMDs were categorized based on their genetic inheritance patterns into either autosomal dominant or autosomal recessive types [58]. Currently, more than 30 distinct genetic forms of LGMD have been categorized (reviewed in [58]), with each form associated with mutations in different genes that impact a range of cellular and molecular pathways essential for muscle function.

Sarcoglycans, a set of transmembrane proteins, are essential for the dystrophin-glycoprotein complex and play a critical role in maintaining the integrity of muscle cell membranes. The loss of δ-sarcoglycan in *Drosophila*, which is related to both mammalian γ- and δ-sarcoglycans, results in a dilated and poorly contractile heart tube, characterized by significantly enlarged end systolic and end diastolic diameters, and progressive impaired locomotion, characterized by shortened sarcomeres and disorganized M lines [59]. The muscle degenerative phenotype in the *Drosophila* mirrors the reduced mobility observed in humans with sarcoglycan mutations. Additionally, mutations in the *TRIM32* gene, which encodes an E3 ubiquitin ligase and are associated with limb-girdle muscular dystrophy (LGMD), have been shown to lead to the abnormal accumulation of certain costamere proteins, including βPS integrin and δ-Sarcoglycan in *Drosophila* models [60,61], suggesting that TRIM32 plays a crucial role in modulating the levels of sarcomeric proteins, thereby preventing the progression of LGMD. The abba mutant, a member of the TRIM/RBCC protein family, exhibits a phenotype resembling that of TRIM32-deficient mice and individuals with LGMD2H who carry TRIM32 mutations. In the absence of abba, there is a significant disruption of F-actin and myosin striations during larval muscle growth. Additionally, abba is crucial for maintaining the structural integrity and function of Z-discs and M-lines in the muscles of adult *Drosophila* [62]. Nevertheless, further investigative efforts are necessary to elucidate the specific ubiquitination patterns mediated by TRIM32 and to determine the ultimate consequences for the targeted substrate proteins.

Mutations in RNPs (ribonucleoproteins) have recently been characterized to play a causal role in MD, suggesting that RNPs are involved in the degeneration of muscle tissue. The expression of the Hrb98DE (the *Drosophila* homolog of hnRNPA2B1) D302V mutant in *Drosophila* leads to a significant accumulation of protein in the cytoplasm, which recapitulates the muscle pathology observed in MDs. The expression of MRJ, which is the *Drosophila* homolog of DNAJB6, inhibits the formation of cytoplasmic inclusions and partially restores the abnormal wing posture phenotype in the Hrb98DE/hnRNPA2 mutants [63]. Thus, any intervention that prevents the irreversible aggregation of disease-associated proteins could potentially reduce their toxicity.

Moreover, mutations in genes related to RNA metabolism and associated with RNA splicing changes, which serve as a molecular signature for MDs, have been described. The *Snupn* gene encodes Snurportin, a protein essential for the nuclear transport of snRNPs, which are key components of the spliceosome. Snurportin variants lead to snRNP accumulation in the cytoplasm of patient-derived fibroblasts and muscle, without affecting the overall expression, indicating the crucial role of Snurportin in snRNP biogenesis [64]. Muscle-specific knockdown *Snup*, the orthologue of human *SNUPN* in *Drosophila*, results in reduction in locomotor capacity and decreased lifespan, both of which are phenotypes associated with muscle development and function [64]. It will be of interest to investigate molecular mechanisms that lead to the formation of protein aggregates associated with diseases, using *Drosophila* as a model organism.

POGLUT1, a key enzyme in the Notch signaling pathway, is responsible for the glycosylation of Notch receptors. Mutations in the *POGLUT1* gene cause muscle development abnormalities in human similar to those observed in *Drosophila* POGLUT1 deficiency, suggesting that POGLUT1 mutations may lead to muscle atrophy and LGMD by affecting satellite cell proliferation and differentiation [65]. *PYROXD1* and HMGCR were identified as pathogenic mutations in LGMD [66,67,68]. Knocking down PYROXD1 and HMGCR ubiquitously in *Drosophila* resulted in lethality, highlighting the critical role of these genes in skeletal muscle development and the pathogenesis of LGMD [66,69]. Together, *Drosophila* provides an effective model for studying LGMD associated mutations. Nevertheless, the relatively simple structure of *Drosophila* muscles means that additional studies in mammalian models are required to confirm these findings.

### 3.5. Congenital Muscular Dystrophy

Congenital muscular dystrophy (CMD) is a hereditary muscular degenerative disease that is considered as congenital glycosylation disorder. The congenital glycosylation disorders within this group exhibit varying phenotypes, ranging from the extremely severe Walker–Warburg Syndrome (WWS) to less severe muscle–eye–brain disease (MEB) and Fukuyama congenital muscular dystrophy (FCMD) [70]. Dystroglycan is highly glycosylated, with its characteristic O-mannosylation being essential for the interaction between dystrophin and the muscle cell’s extracellular matrix [12,26]. Therefore, CMD may be associated with mutations in either dystroglycan itself or in the O-linked glycosyltransferases that are responsible for its glycosylation.

WWS is a severe form of CMD, often accompanied by brain malformations, which is caused by mutations in the *O-mannosyltransferase* 1 and *O-mannosyltransferase* 2 (*POMT1* and *POMT2*) gene [71]. Defects in the *POMT1* gene lead to glycosylation defects in dystroglycan, impairing its ability to interact properly with basement membrane proteins, such as laminin. WWS patients typically exhibit severe muscle weakness, delayed motor development, and intellectual disabilities shortly after birth. The disease progresses rapidly, often resulting in early death. WWS has been reproduced by mutating the *Drosophila* orthologs of *POMT1* and *POMT2*, named *rotated abdomen* (*rt*) and *twisted* (*tw*), respectively [23,72]. The *rt* and *tw* mutant *Drosophila* exhibit muscle defects and phenotypes resembling those associated with reduced dystroglycan function. These mutants show thin or deficient muscles in the abdomen, muscle attachment, and contraction issues and structural muscle abnormalities such as sarcomeric disarray and swollen organelles. The mutants’ climbing and flying abilities decline with age, similar to symptoms in WWS patients. The excessive apoptosis of myoblasts and failure to reduce myoblast numbers in wing imaginal discs are observed in *rt* mutants [23,30,73]. The absence of COL4A1 (type IV collagen) also leads to WWS. In the *Drosophila col4a1* mutants, the loss of sarcomere structure and the disintegration and streaming of Z-discs, as well as irregular actin aggregation, atrophy, and abnormal fiber size, were observed [31].

### 3.6. Emery–Dreifuss Muscular Dystrophy

Emery–Dreifuss muscular dystrophy (EDMD) is a rare form of muscular dystrophy, and its early diagnosis is crucial because of the potential for life-threatening cardiac complications. Typically, EDMD is characterized by muscle weakness, early contractures, cardiac conduction defects, and cardiomyopathy, though the presence and severity of these symptoms vary depending on the subtype and individual [74]. The genes associated with EDMD include *EMD*, *LMNA*, *SYNE1*, *SYNE2*, *FHL1*, *TMEM43*, *SUN1*, *SUN2*, and *TTN*, which encode the proteins Emerin, Lamin A/C, Nesprin-1, Nesprin-2, FHL1, LUMA, SUN1, SUN2, and Titin, respectively [75].

In *Drosophila*, the *lamin C* gene is homologous to the human A-type *Lamin* gene. The expression of truncated Lamin C (lacking the first 42 amino acids or the N-terminal head domain) in *Drosophila* larval muscles results in muscle defects and nuclear envelope alterations, which further lead to transcriptional repression and gene expression inhibition [76]. In addition, *lamin C* mutations lead to the loss of βFtz-F1 expression during the pupal stage, limiting normal muscle contraction and subsequently affecting muscle morphology and function [76]. *Lamin C* mutations also cause defects in tendon cells and the severe fragmentation of muscle cell nuclei [77]. MSP-300, a *Drosophila* homolog of nesprin, locates at the Z-line of the sarcomere. Msp-300^ΔKASH^ mutants exhibit impaired locomotion and reduced GluRIIA receptor density at the *Drosophila* neuromuscular junction (NMJ), further affecting synaptic signaling and muscle function [78].

Moreover, genes such as *Otefin*, *bocksbeutel*, *klaroid*, and *klarsicht* are required for the separation and correcting position of nuclei in *Drosophila* [43]. These genes correspond to the Emerin, Nesprin, and SUN protein families and interact with the nuclear envelope to ensure the integrity of the nuclear membrane and the proper localization of nuclear materials [79,80]. Mutations in these genes, particularly in muscle and tendon cells, lead to nuclear mispositioning, muscle morphology abnormalities, and functional defects.

### 3.7. Oculopharyngeal Muscular Dystrophy

Oculopharyngeal muscular dystrophy (OPMD) is an autosomal dominant genetic disorder characterized by the progressive degeneration of specific muscles. OPMD is caused by mutations in the gene encoding poly(A)-binding protein nuclear 1 (PABPN1), which results in an extension of 11 to 18 alanine residues at the N-terminus of the protein. The extension of the alanine repeat induces misfolding and aggregation of PABPN1 in the muscle cell nuclei [81]. The introduction of PABPN1 with a 17 alanine repeat sequence in *Drosophila* resulted in muscle degeneration, wing position abnormalities, and apoptosis. This was marked by disorganized myofibrils, nuclear inclusions, mitochondrial loss, and vacuolation, all of which are reminiscent of the muscle degeneration observed in patients with OPMD [82]. However, the polyalanine tract is not an absolute necessity for inducing muscle degeneration. The deletion or mutation of RRM, an RNP-type RNA-binding domain of PABPN1, prevented the appearance of the OPMD-like phenotype, even though there are 17 alanine repeats within PABPN1 [82]. These findings suggest that the ability of PABPN1 to bind RNA via the RRM domain is essential for the pathogenesis of OPMD [83]. The *Drosophila* model serves as an excellent genetic counterpart to transgenic mice in the study of muscular dystrophies.

Endoplasmic reticulum (ER) stress activates the unfolded protein response (UPR) in the *PABPN1* mutant *Drosophila*, leading to the muscle degeneration and accumulation of mutant PABPN1 protein in muscle cells. Additionally, mutations in the key genes of the PERK branch of the UPR alleviate muscle degeneration and PABPN1 aggregation, thereby confirming the crucial role of endoplasmic reticulum stress in the pathogenesis of OPMD [81]. Additionally, the loss of PABPN1 also leads to an increased activity of the ubiquitin–proteasome system, which may be associated with the degradation of muscular proteins [84].

Our review of *Drosophila* research in the context of MD highlights the significant contributions of *Drosophila* models in elucidating molecular mechanisms and discovering potential therapeutic targets. Table 1 summarizes the principal phenotypes in *Drosophila* models caused by genes linked with various MD, along with their corresponding human pathologies. This review aims to establish a foundation for understanding the importance of *Drosophila* in MD research and to facilitate the translation of these findings from bench to bedside.

## 4. Conclusions

*Drosophila*, used as a powerful model organism for human disease, shares highly conserved genetic and functional features with humans. The essential genes associated with muscular dystrophies, including the genes encoding dystroglycan and glycosyltransferases, have homologs in *Drosophila*. The simple muscle system and rich genetic tools available in *Drosophila* make it an ideal platform for investigating the molecular and cellular mechanisms of muscular dystrophy. Advances in genomics and molecular biology have greatly enhanced our understanding of the underlying causes of muscular dystrophy. Mutations in key genes that are related to muscle membrane stability and protein glycosylation disrupt the interaction between muscle cells and the extracellular matrix and then lead to a range of muscle defects.

In particular, studies on *Drosophila* have revealed critical molecular pathways and functional characteristics associated with the disease-causing gene, not only enhancing our understanding of muscular dystrophy pathogenesis but also providing valuable insights for potential therapeutic strategies. *Drosophila* serves as a potent tool for genetic manipulation and disease modeling, yet its physiological differences from mammals limit its ability to precisely mirror human MD pathology. Therefore, integrating findings from *Drosophila* with those from mammalian models is essential for gaining a comprehensive understanding of disease mechanisms. Meanwhile, there are many pathogenic genes identified in mammals that have not been validated or discovered in *Drosophila*, which limits the full application of *Drosophila* as a model organism for muscular dystrophy. Furthermore, it will be of interest to investigate the evolutionarily conserved mechanisms among muscular dystrophy by utilizing state-of-the-art *Drosophila* tools and advanced molecular biology techniques, offering to develop more effective treatments for these debilitating diseases.

*Drosophila* models are highly effective for screening large libraries of compounds to assess the effects of potential drugs on muscle strength and degeneration. By using *Drosophila* mutants that mimic human MD, we can identify compounds that slow the rate of muscle degeneration and enhance muscle performance. Moreover, employing *Drosophila* “avatars” to mirror the genetic profiles of patients represents a novel approach in personalized medicine. By creating flies with the exact mutations found in patients, we can screen various therapeutic agents to determine the most effective treatment for each individual’s specific mutation. By leveraging the *Drosophila* models of MD, along with advanced tools such as CRISPR and high-throughput targeted drug screening, *Drosophila* offers an invaluable platform for both basic research and clinical applications (Figure 2). These models enable the discovery of new therapeutic targets, the validation of existing drugs, and the customization of treatments to suit individual genetic profiles.

## Figures and Tables

**Figure 1 ijms-26-01459-f001:**
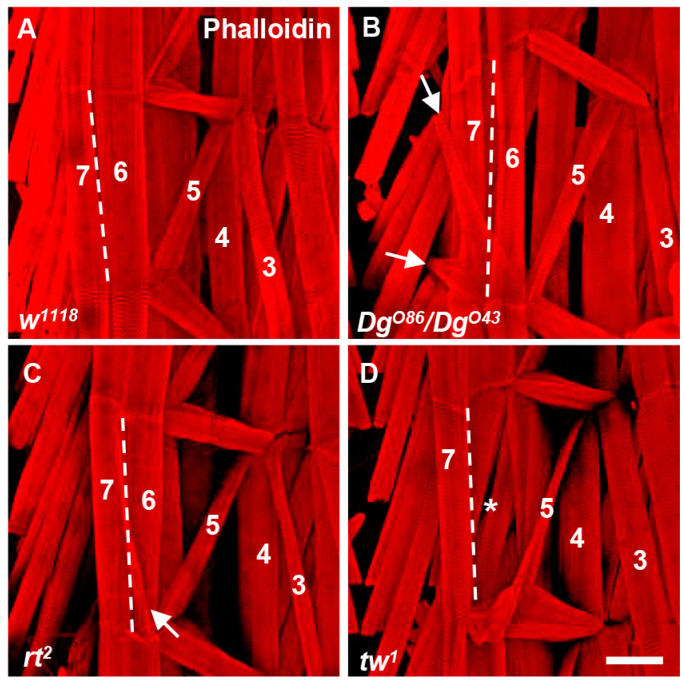
The muscle attachment defects caused by mutation of MD associated with genes in *Drosophila*. Phalloidin-stained muscles of *Drosophila* larvae showing muscle attachment defects in the indicated genotypes. The muscles referred to in the text are labeled by number according to standard nomenclature. The dashed lines represent the boundaries of muscles 6 and 7. (**A**) The arrangement of muscles in part of a hemisegment in w1118 files. (**B**,**C**) The arrow indicates muscle that is abnormal and should not be present, suggesting that muscle proliferated excessively. (**D**) The asterisk represents muscle loss. Scale bar, 100 μm.

**Figure 2 ijms-26-01459-f002:**
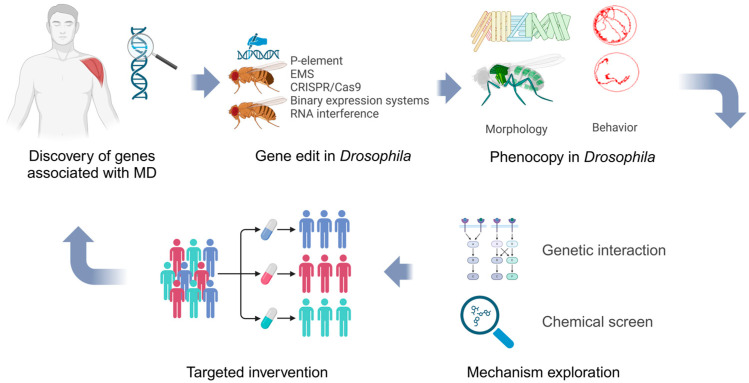
*Drosophila melanogaster* serves as a suitable platform for both basic scientific research and clinical applications.

**Table 1 ijms-26-01459-t001:** *Drosophila* models for muscular dystrophy caused by associated genes.

Disease Type	Modeling in *Drosophila*	Pathology in Human	Key Genes
Duchenne and Becker muscular dystrophy (DMD/BMD)	Muscle rupture, detachment from tendon cells, progressive climbing defects, reduced lifespan, and cardiac abnormalities	Muscle degeneration, progressive mobility loss, and dilated cardiomyopathy	DMD (encodes the dystrophin protein, all isoforms or Dp117) [39,40]
Myotonic dystrophy (DM)	RNA toxicity, RNA foci formation, sarcomeric protein dysregulation, and muscle degeneration	RNA foci in cells, muscle weakness, conduction defects, and arrhythmias	MBNL [44,45,46], CUGBP1 [46], and CNBP (related to DM2) [85]
Facioscapulohumeral muscular dystrophy (FSHD)	Winged scapula-like phenotype and muscle degeneration	Facial and proximal arm weakness and asymmetric muscle involvement	DUX4 [54] and FRG1 [54]
Limb-girdle muscular dystrophy (LGMD)	Shortened sarcomeres, disorganized M lines, impaired flight ability, and Z-disk disorganization	Pelvic and shoulder girdle muscle weakness and progressive atrophy	abba [62] and TNPO3 [86]
Congenital muscular dystrophy (CMD)	Thin or deficient abdominal muscles, attachment defects, sarcomeric disarray, and increased apoptosis of myoblasts	Severe muscle weakness at birth and brain abnormalities (e.g., Walker–Warburg Syndrome and Fukuyama CMD)	POMT1 [30], POMT2 [30], COL4A1 [31], and Lamin C [76]
Emery–Dreifuss muscular dystrophy (EDMD)	Muscle and tendon defects, nuclear envelope fragmentation, and impaired nuclear positioning	Early contractures, cardiac conduction defects, and progressive muscle weakness	Emerin [79] and Nesprin [79]
Oculopharyngeal muscular dystrophy (OPMD)	Muscle degeneration, nuclear inclusions, disorganized myofibrils, and ER stress activation	Progressive muscle weakness in eyelids, throat, and proximal limbs	PABPN1 [81]

## Data Availability

No new data were created or analyzed in this review. BioRender was used to create and edit the scientific illustrations (Figure 2, created in BioRender. GENG, J. (2025) https://BioRender.com/w34f363 (accessed on 6 February 2025)) used in this work.

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
