# Peer review of "Drosophila melanogaster: A Model Organism in Muscular Dystrophy Studies"

_ijms, 2025, doi:10.3390/ijms26041459_

Round 1

Reviewer 1 Report

Comments and Suggestions for Authors

This review by Zhao et al., highlights the use of Drosophila as a versatile model organism for studying muscular dystrophy, offering valuable insights into genetic manipulations, disease modeling, molecular pathways, and potential therapeutic targets. The review is comprehensive and well written. There are a few concerns-

1. The author list appears incomplete.

2. It would be informative if the authors could include a schematic diagram showing the different forms of MD and kind of provide a graphical summary of the paper even though the table does a good job.

3. In the section "The Advantages of Drosophila in MD Research", the authors should include detailed examples of how the genetic tools have been used in Drosophila in the context of MD.

4. It will be informative if the authors when describing the different forms of MD in flies can include details about how the phenotypes observed in flies correlates to the phenotypes observed in humans. Some of this information is included in the table, but it should also be added to the text.

5. The line 112-113: "Drosophila models expressing expanded repeat RNA sequences toxicity linked to DM1 or DM2 [17,40,41]." doesn't make sense

6. The authors should include a section that describes in details with examples how Drosophila can be used/is currently being used as a system to identify therapeutic targets for MD.

Author Response

Thank you for your comments and suggestions. We have modified the text to address your concerns (text highlighted in blue).  Please see our detailed point-by-point response below.

Comment 1: The author list appears incomplete.

Response 1: We apologize for the oversight. All authors had been listed properly in the manuscript. We filled in the first two authors in the system for submission, and all authors had read and agreed to submit the manuscript.

Comment 2: It would be informative if the authors could include a schematic diagram showing the different forms of MD and kind of provide a graphical summary of the paper even though the table does a good job.

Response 2: Thank you for the suggestion. Due to the complexity of clinical manifestations and genetic heterogeneity, we compared the different forms of MD in Table 1, rather than summarizing them together in a graph. We have now included Figure 1 to show the different types of defects in Drosophila larvae muscles based on our previous study (lines 80-90). Additionally, we have also added Figure 2 to illustrate Drosophila can be used for both basic scientific research and clinical applications (lines 408-10). We hope this will make it easier for readers to understand.

Comment 3: In the section "The Advantages of Drosophila in MD Research", the authors should include detailed examples of how the genetic tools have been used in Drosophila in the context of MD.

Response 3: Thank you for bringing up this question. In lines 91-113, we have now included several examples of how genetic tools have been used in Drosophila to study muscular dystrophy (MD), with these examples highlighted in blue for clarity.

Comment 4: It will be informative if the authors when describing the different forms of MD in flies can include details about how the phenotypes observed in flies correlates to the phenotypes observed in humans. Some of this information is included in the table, but it should also be added to the text.

Response 4: We appreciate the suggestion. Not all Drosophila mutated in MD associated genes can mimic the phenotypes observed in DM patients. We compared the phenotypes observed in flies that correlate with those observed in humans, such as in the lines 135-39, 226-32, 238-40, 249-55, 272-76, 304-10, and 353-57. To make the information more readable, we have summarized the detailed comparable phenotypes in Table 1 based on previous studies. The phenotypes observed in Drosophila are summarized in the section titled "Modeling in Drosophila," and the corresponding phenotypes in humans are detailed in the section titled "Pathology in Humans." This comparable information has been included in the manuscript.

Comment 5: The line 112-113: "Drosophila models expressing expanded repeat RNA sequences toxicity linked to DM1 or DM2 [17,40,41]." doesn't make sense

Response 5: We have rephrased the sentence as following: “Noncoding microsatellite repeat expansions, such as (CTG) expansions in DMPK and (CCTG) expansions in CNBP, are examples of toxic RNA gain-of-function mechanism in MD, underscoring that splicing defects are directly linked to the pathology of MD”. (Lines 155-59)

Comment 6: The authors should include a section that describes in details with examples how Drosophila can be used/is currently being used as a system to identify therapeutic targets for MD.

Response 6: Thank you for bringing up this question. We have added a section to discuss how Drosophila can be used as a system to identify therapeutic targets for MD in lines 411-424.

Reviewer 2 Report

Comments and Suggestions for Authors

The manuscript from Zhoa et al., summarizes very nice the known literature about human related muscles dystrophies in the model organism Drosophila. The review highlights the usage of the model organism to study these muscle abnormalities.

The manuscript is well written and I strongly recommend for publication, but I have some pointes that should be addressed:

1.     The authors write in the introduction: “The lack of these molecules disrupt the interactions between muscle cells and the extracellular matrix, resulting in the defects on muscle development and pathological changes.“ I found that a bit one sided, since the authors also address the LGMD, which also contains changes within the muscle structure stability and not only the connection between muscle and excellurlar matrix.

2.     In the LGMD section the authors are missing one publication of Domsch et al., 2013, which points to a function of Drosophila tn/abba (TRIM32) to M-line and Z-disc stability.

3.     Maybe the authors could write out Mbl and Bru3 in the sentence:” These foci co-localized with Mbl and were accompanied by 117 stable expression of CELF1/Bru3, mimicking the characteristics observed in DM1 patient.” If Mbl means muscle blind.

Author Response

Thank you for bringing up this question. We have significantly modified the manuscript to address your concerns (text highlighted in blue), so it is accessible to a broader readership. Please find the detailed point-by-point response below.

Comment 1: The authors write in the introduction: “The lack of these molecules disrupt the interactions between muscle cells and the extracellular matrix, resulting in the defects on muscle development and pathological changes.“ I found that a bit one sided, since the authors also address the LGMD, which also contains changes within the muscle structure stability and not only the connection between muscle and excellurlar matrix.

Response 1: We apologize for the oversight. We thought that muscle structure unstable would be the result of muscle development and pathological changes, so mixed them in the manuscript. We have now rephrased the sentence as “The lack of these molecules disrupts the interactions between muscle cells and the extracellular matrix, resulting in defects in muscle development, pathological changes, and instability in muscle structure.”, which has been added in lines 43-46.

Comment 2: In the LGMD section the authors are missing one publication of Domsch et al., 2013, which points to a function of Drosophila tn/abba (TRIM32) to M-line and Z-disc stability.

Response 2: We apologize for the oversight. The reference has been included properly in lines 238-44.

Comment 3: Maybe the authors could write out Mbl and Bru3 in the sentence:” These foci co-localized with Mbl and were accompanied by 117 stable expression of CELF1/Bru3, mimicking the characteristics observed in DM1 patient.” If Mbl means muscle blind.

Response 3: Thank you for the suggestion. Bruno 3 (Bru3) is the ortholog of CELF1 in Drosophila[1]. We have rephrased the sentence as following: “These foci co-localized with Muscleblind (Mbl) and were accompanied by the stable expression of CELF1 in Drosophila, mimicking the characteristics observed in DM1 patients.” (Lines 163-66)

  1. Souidi, A.; Nakamori, M.; Zmojdzian, M.; Jagla, T.; Renaud, Y.; Jagla, K. Deregulations of miR-1 and its target Multiplexin promote dilated cardiomyopathy associated with myotonic dystrophy type 1. EMBO Rep 2023, 24, e56616